# `PeskAAS`: A near-real-time, open-source monitoring and analytics system for small-scale fisheries

**Alexander Tilley** [1]☯*, **Joctan Dos Reis Lopes** [1], **Shaun P. Wilkinson** [2,3]☯

**1** WorldFish, Bayan Lepas, Penang, Malaysia, **2** School of Biological Sciences, Victoria University of Wellington, Wellington, New Zealand, **3** Wilderlab, Wellington, New Zealand

☯ These authors contributed equally to this work.
* alex.tilley@gmail.com

**Data Availability Statement:** The peskAAS catch questionnaire, R script and data used to test the application in this study are archived in Dataverse https://doi.org/10.7910/DVN/TVGIJJ. The peskAAS

## Abstract

Small-scale fisheries are responsible for landing half of the world's fish catch, yet there are very sparse data on these fishing activities and associated fisheries production in time and space. Fisheries-dependent data underpin scientific guidance of management and conservation of fisheries systems, but it is inherently difficult to generate robust and comprehensive data for small-scale fisheries, particularly given their dispersed and diverse nature. In tackling this challenge, we use open source software components including the *Shiny* R package to build `PeskAAS`; an adaptable and scalable digital application that enables the collation, classification, analysis and visualisation of small-scale fisheries catch and effort data. We piloted and refined this system in Timor-Leste; a small island developing nation. The features that make `PeskAAS` fit for purpose are that it is: (i) fully open-source and free to use (ii) component-based, flexible and able to integrate vessel tracking data with catch records; (iii) able to perform spatial and temporal filtering of fishing productivity by fishing method and habitat; (iv) integrated with species-specific length-weight parameters from *FishBase*; (v) controlled through a click-button dashboard, that was co-designed with fisheries scientists and government managers, that enables easy to read data summaries and interpretation of context-specific fisheries data. With limited training and code adaptation, the `PeskAAS` workflow has been used as a framework on which to build and adapt systematic, standardised data collection for small-scale fisheries in other contexts. Automated analytics of these data can provide fishers, managers and researchers with insights into a fisher's experience of fishing efforts, fisheries status, catch rates, economic efficiency and geographic preferences and limits that can potentially guide management and livelihood investments.

## Introduction

Approximately half of the global catch of fish is landed in small-scale fisheries (SSF) [1], yet the ability for science to guide the sustainable exploitation of these resources is inhibited by

code and documentation are hosted at https://github.com/shaunpwilkinson/peskAAS.

**Funding:** This work was undertaken as part of the CGIAR Research Program on Fish Agri-Food Systems (FISH) led by WorldFish, and the CGIAR Big Data Platform Inspire Challenge 2018 led by CIAT and IFPRI. These programs are supported by contributors to the CGIAR Trust Fund. PeskAAS was established under the Fisheries Sector Support Program funded by the Royal Norwegian Embassy in Jakarta. The funders provided support in the form of salary for authors [A.T.] and field costs, but did not have any additional role in the study design, data collection and analysis, decision to publish, or preparation of the manuscript. Wilderlab is a recent, unassociated affiliation for S.P.W. and did not contribute funding towards the study. The specific roles of these authors are articulated in the 'author contributions' section.

**Competing interests:** The authors declare there are no competing interests. SPW's affiliation to Wilderlab does not alter our adherence to PLOS ONE policies on sharing data and materials.

substantial data gaps. There are 40 million people actively fishing in inland and coastal environments worldwide, yet due to the challenges involved in collecting data in dispersed, informal and diverse fisheries, the contributions of SSF to livelihoods, and food and nutrition security, are frequently hidden and poorly accounted for [2]. As a result, fisheries solutions are underrepresented in investments and policies to address food and nutrition security [3]. Recent research has shown that the nutrient qualities and quantities of fish already caught would be sufficient to address major micronutrient deficiencies for many countries [4], and that fish-based food strategies hold untapped potential to address nutrition short falls.

Refinements to the management of fisheries towards sustainability and food and nutrition security goals can be supported, amongst other things, by higher quality and more comprehensive fisheries and value chain data. Addressing these gaps with small-scale fisheries in low-income countries requires sensitivity to the limits of resources and capacity to collect, store, analyse and respond to these data [5]. Even with adequate data, much fisheries science relies on statistical models and analytics that were developed for single species fisheries, rather than biodiverse tropical fisheries where multiple species frequently comprise the harvests from diverse and biodiverse habitats [6]. Many diagnostic and statistical tools have been developed to support fisheries decision-making or stock assessment in data-poor contexts, e.g., FishPath [7] or TropFishR [8]. However, very few solutions provide end-to-end integration from data collection at the point of fishing or landing, to visualisation of data summaries for managers and fishing communities. This leads to a situation where management and policies are built on poor government estimates or reconstructed trade and consumption statistics that are often not relevant at the local scale, and hence are highly susceptible to governance and management failures [9].

Over the past two decades, the proliferation of information and communication technologies (ICTs) has revolutionised the collection, communication and storage of data in food production sectors, including in the industrial fishing sector [10]. These ICTs innovations commonly include design and development of mobile smartphone applications and digital survey forms [11]. There are some examples of applications and tools to collect fisheries data on a small scale, the most successful of which are generally 'high touch', meaning they involve significant contextual development that cannot easily be scaled to other systems or geographies (see [12–15]). The danger in developing scalable technologies is that they are often imposed as prescribed 'solutions' on low-income countries, and can merely reinforce the capacity gap, alienate managers and stakeholders, and be ill-suited to the contextual reality [16, 17]. There is an urgent need for a light touch, scalable and integrated approach to data collection and analytics, but thus far, this has been elusive in terms of getting simple, usable data in the hands of fisheries managers.

Here, we describe a new digital application called PeskAAS—a pseudo-acronym for fisheries (*peskas*) in the national language of Timor-Leste, Tetum, combined with Automated Analytics System. PeskAAS connects open source programs to collect, communicate, analyse and visualise SSF movement and catch data. We piloted PeskAAS to determine how scripted analytics can assist fisheries management by generating digestible, summarised information rapidly for decision-making in data-deficient and low capacity systems. We provide examples to illustrate the types of analytics and machine learning that the system can be used to carry out. Our test site of Timor-Leste was chosen due to the very limited level of fisheries development, the paucity of information, and the limited fisheries governance capacity.

## PeskAAS overview

PeskAAS is an interactive web-hosted R *Shiny* application developed to facilitate data exploration and decision-making processes in small-scale fisheries. This application accesses the 'peskaDAT' database in real-time using the DBI and RMySQL R packages [18, 19], and pulls

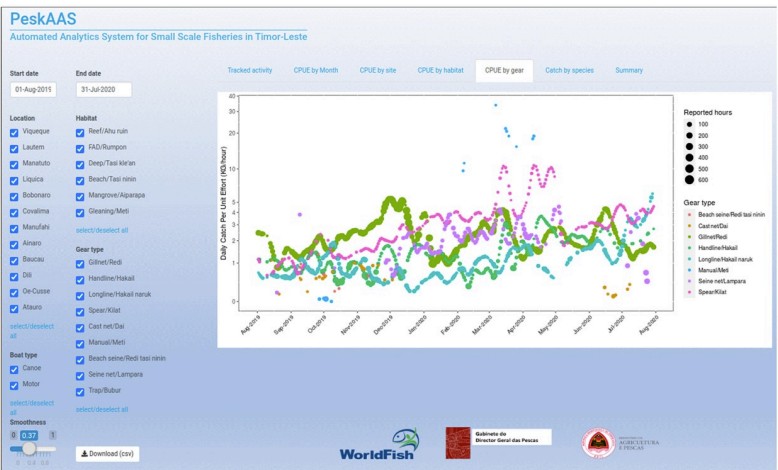

**Fig 1. Screenshot from the interactive `PeskAAS` dashboard developed to visualise near-real time fish landings data in Timor-Leste.** https://worldfish.shinyapps.io/PeskAAS/.

catch records collected by enumerators at landing sites into a web-based interactive R session (hosted by https://www.shinyapps.io). From this dashboard, users can apply dynamic location, gear, habitat and date-range filters to create informative plots for catch per unit effort (CPUE), total catch, species composition and total national catch estimates (Fig 1).

The user interface incorporates tick-box widgets for dynamic selection of municipalities, fishing habitats, gear types and boat type, and a slider widget is included for users to fit cubic smoothing splines weighted by trip effort to the CPUE data with a user defined smoothing parameter that is dynamically adjustable to aid in visualization. Users also have an option to download a month-aggregated summary table in .csv format for further analysis, using a download-button widget.

The `PeskAAS` application is hosted remotely by shinyapps.io (https://www.shinyapps.io/) on the free pricing tier, which currently allows 25 active hours of use per month. Users can also run the application locally, by cloning the repository from GitHub and launching the application from a local R session, or by installing RStudio Shiny Server (https://www.rstudio.com/products/shiny/shiny-server/) and hosting the application on a local machine.

The `PeskAAS` application consists of a network of freely available components for data collection, storage, manipulation and analysis, catering to humanitarian organizations with low usage requirements (< 25 active hours per month) (Fig 2). The application is scalable for higher usage levels with modest subscription levies for KoBo toolbox, Shiny (https://www.shinyapps.io), Heliohost (https://www.heliohost.org) and Google Cloud Platform (GCP; https://cloud.google.com).

## Methods

We developed and piloted PeskAAS in Timor-Leste from 2016 to 2019. In May 2019, the Timorese Government adopted and launched this technology as their official national fisheries monitoring system.

### Catch documentation at landing sites

Fisheries catch data were collected on 3G-enabled android tablets using a digital survey form developed in KoBo toolbox (http://www.kobotoolbox.org/) a free suite of tools for field data

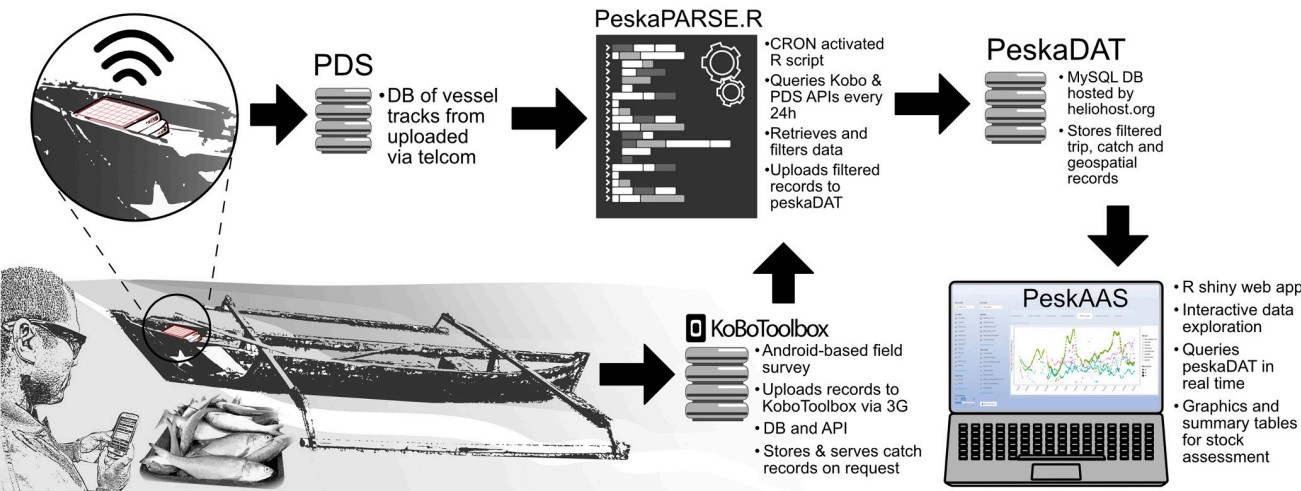

**Fig 2. A diagrammatic representation of the `PeskAAS` application.** From bottom left, the catch from a fishing vessel is entered into a KoboCollect survey form on a smartphone. These data are uploaded to the KoboToolbox database (DB). The PesksPARSE.R script pulls these data along with the movement track of the fishing trip from PDS via an application-programming interface (API) every 24 hours. These data are then checked and filtered, and uploaded to the PeskaDAT database. The `PeskAAS` Shiny web app queries PeskaDAT and displays near-real-time graphics and analytics.

collection. One data enumerator was hired in each of the 11 coastal districts to cover 25 landing sites across Timor-Leste (Fig 3). Atauro Island to the north of Dili (the national administrative centre) falls within the municipality of Dili, and is a disproportionately important area for national fisheries with a high population of fishers, so three enumerators are responsible to record landings across 12 small landing sites. Enumerators interview individual fishers (i.e., the names of fishers are collected, and their catch profile directly associated to them) when they return from fishing. Data collected include transport type (motor, canoe, on foot), gear type (gill net, seine net, beach seine, hand line, long line, spearfishing, trap, gleaning), habitat type where fishing took place (reef, fish aggregating device, deep, shore or mangrove), number of fishers (i.e., number of men, women and children engaged in that fishing trip), trip duration (rounded to nearest hour), species or species group, sizes (fork length to nearest cm) of all fish landed and number caught by species or species group, and the sale price per kg or per number of fish, if the catch was destined for market. The catch survey form evolved through time to integrate a growing list of fishers and identified species, and in response to various feedback mechanisms, such as error flagging and workshops with fisheries managers. Species of

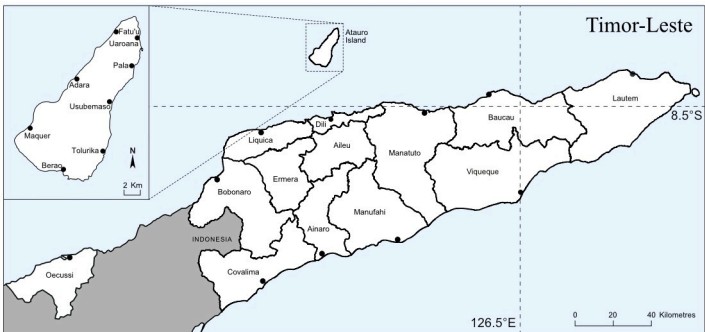

**Fig 3. Map of Timor-Leste illustrating landing sites where small-scale fisheries catch was recorded using smartphones or tablets.**

particular local importance were selected based on index of relative importance analysis on Timor-Leste fisheries [20] and these are retained at species level. All other species captured in the fishery are classified by family, or aggregated further into one of the nine groups of species constituting the Marine Fishes division of the FAO International Standard Statistical Classification of Aquatic Animals and Plants [21] (S1 Data).

All research was authorised and conducted in partnership with the Timor-Leste Ministry of Agriculture and Fisheries. All catch data collection was carried out with the verbal consent of the fisher. All participation in catch data monitoring and vessel tracking was voluntary, and fishers not willing to participate were not prejudiced against in any way.

## peskaDAT database

We developed a cloud-based MySQL database of filtered landings records called 'peskaDAT', to host cleaned and checked fisheries landings data, geo-located boat tracks and ancillary tables such as species, boat, gear and habitat information in an easily accessible and mutable format (S2 Data). This database is hosted by Heliohost, a free hosting platform with several MySQL relational database servers. A monthly database backup is created and stored locally in.rds format (an R binary data file) in a Dropbox folder to ensure data perpetuity and backwards compatibility.

## peskaPARSE.R script

To automatically access landing records from KoBo toolbox, and filter, manipulate and submit the cleaned records to the peskaDAT database on a regular (daily) basis, we developed an R script called 'peskaPARSE.R' (S3 Data) that is scheduled on a daily cron job (a command to a server to be executed at a specified time) run on a Google Cloud Platform virtual machine. The script first accesses the KoBo toolbox API (application programming interface that allows access to the data) and pulls the new catch records into the R environment. The catch weight for each new record is estimated using length-weight parameters obtained from FishBase (www.fishbase.org) using the rfishbase R package [22]. Several filters are applied according to thresholds including species- or group-specific length, number of individuals, weight and price boundary checks, formatting error checks, and invalid gear/habitat combination checks. Suspect entries are flagged for curation by a moderator, and their values are automatically suppressed from contributing to downstream applications (including the PeskAAS dashboard in Fig 1) until records are amended in the KoBo toolbox dashboard. This is achieved via prompt follow-up clarification with field-based observers, to ascertain whether data entry errors were made during form submission or whether the record in question is a genuine outlier.

To automatically run the peskaPARSE.R script on a daily basis, we deployed a free Ubuntu 18.04 minimal virtual machine (VM) instance on the Google Cloud Platform, transferred the script to a convenient user directory (specified by the [address] argument below), and scheduled a 24-hourly cron job by appending the following line to the /etc/crontab file:

```
0 0 * * * root Rscript -e 'source("[address]")'
```

where [address] is replaced by the web location of the R script. In our case we stored the script in a publicly available Dropbox directory and modified the public link to terminate with download = 1 to enable direct sourcing (see https://github.com/shaunpwilkinson/ PeskAAS for further details). For the script to successfully import the raw trip- and landing records from the KoBo and Pelagic Data Systems APIs, and deposit the filtered records in the peskaDAT MySQL database, three authorization files must be stored in the same directory as the peskaPARSE.R file: a single-line text file named KoBo-auth.txt containing the KoBo toolbox username and password, delimited by a colon operator, and similarly-formatted files

containing log-on credentials for the Pelagic Data Systems dashboard (PDS-auth.txt) and the peskaDAT MySQL database (peskaDB-auth.txt; user with SELECT privileges). The regular cleaning and filtering of new landings records ensures that the data available for analysis and visualization are timely and trustworthy.

### Geospatial vessel tracking

To visualise geospatial fishing effort and extrapolate fishing effort from individual trips to community, municipal and national levels, we installed tamperproof solar-powered GPS units (PDS v1.25c) developed by Pelagic Data Systems Inc. (San Francisco, CA) on to 5–15 boats per landing site (N = 437). Tracking was voluntary and fishers bore no costs. Trackers record point location data automatically every 5 seconds and communicate those data when in range of a cellular network (Fig 4). Vessel tracks were also linked to the trip's catch data, where available, using the unique GPS unit ID, allowing us to train a model to predict unknown variables for trips with GPS data only, such as gear and habitat type.

### Fisheries analytics

To quantify how catch volume varied over time and space, the catch-per-unit-effort (CPUE) is used as a metric for relative fish abundance, and to track the response of fish stocks to fishing pressure [24]. For CPUE, a standard unit of *effort* is required. For `PeskAAS`, we use raw CPUE, with effort standardized into the unit of *fisher-hours* on each fishing trip, calculated as the trip duration in hours multiplied by the number of people who were actively fishing on that boat during that trip. The complexity of more robust CPUE standardization (see [24]) of multiple gear types in tropical, multi-habitat, mixed species fisheries makes it impractical in a livelihoods context, so fisher hours is used as a unit applicable across all fisheries and scales.

To generate accurate regional and national catch and CPUE estimates, it is necessary to estimate the frequency and duration of fishing trips for each boat type, known as a vessel activity coefficient (VAC). To generate an accurate VAC accounting for differences in boat type, vessel tracks were used to generate monthly VAC values according to motorized and non-motorized

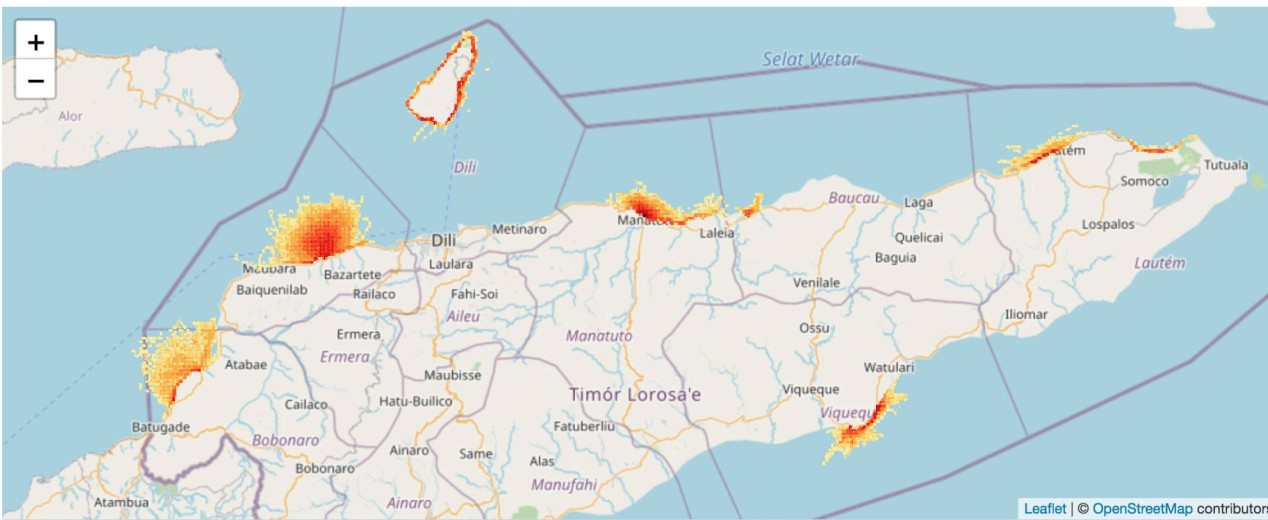

**Fig 4. The relative effort heat map of small-scale fishing vessel tracks between August 2018 and February 2020 in Timor-Leste from the `PeskAAS` dashboard.** Red shading represents areas of highest effort (near to homeports and shore), and orange to yellow shading shows areas fished with decreasing effort further from shore. Map data © OpenStreetMap contributors. Map layer by Esri via leaflet [23].

vessels, and were used to calculate the first national scale estimates of fish production based on SSF landings in Timor-Leste (Eq 1).

The national monthly catch (in tons; C) is estimated as:

$$C = \sum\nolimits_{b} CPUE_b \times EPT_b \times VAC_b \times N_b \times 0.001 \tag{1}$$

where $b$ refers to the boat type (canoe or motor-powered; shore based fishing accounted for $< 2\%$ of reported landings and was hence omitted from the analysis); $CPUE$ is the monthly catch per unit effort (averaged across all sites, in kg per fisher-hour); $EPT$ is the median effort per trip (fisher-hours); $VAC$ is the vessel activity coefficient (average number of trips per month); and $N_b$ is the total number of boats of type $b$ in the national fleet (national fisher and boat census data obtained from the Timor-Leste Ministry of Agriculture and Fisheries).

### Supervised prediction of missing trip attributes

The integration of shore-based observer data with high-resolution geo-located vessel fishing tracks from on-board GPS units from *Pelagic Data Systems* Inc. allowed us to validate a supervised classification approach for predicting gear and habitat types in the case where only GPS data are available (i.e. trips that are tracked but catch is not recorded by enumerators). For each tracked but unobserved trip, a donor trip is selected from the subset of trips for which both GPS and observer data are available, based on spatial nearest-neighbour analysis [25]. To achieve this, ten GPS vectors from each trip spaced at even time intervals are used to generate training- and query-data matrices for input into the nn2 function in the RANN R package [26]. Donor trips are assigned based on minimum Euclidean distance and are only assigned if the distance is $< 0.20$ km. To validate this approach, we randomly allocated the linked trip set available as of 31 July to 80% training data and 20% query data, and tested the success rate in terms of number of gear and habitat types correctly predicted in the query set over 100 iterations. The nearest-neighbour classification approach correctly predicted 83.4% (SD = 0.023%) of gear types and 91.9% (SD = 0.021%) of habitat types based on 741 training observations and 183 query observations.

### Results & discussion

At the pilot's inception in 2016, the Timor-Leste government and fishers had very sparse quantitative data with which to characterise fisheries or determine production trends over time. PeskAAS was developed concurrently with the co-generation of a national database of SSF landings with the Timor-Leste National Fisheries Directorate. As of 31 July 2020, 59,000 fishing trips have been geotracked, representing 810,000 km travelled over 313,000 hours. Catch data was recorded for 29,251 trips, resulting in 32,872 landings records representing a total catch weight of 298.5 tonnes. `PeskAAS` has already been utilised to test the effectiveness of fish aggregating devices at the boosting catch rates of nearshore artisanal fishers [27] and enabled the first calculation of national fisheries production for the Timor-Leste SSF fleet [20]. The full adoption of `PeskAAS` as the official national fisheries monitoring system of the Timorese government in 2019 suggests legitimacy and the potential for sustainable fisheries on a national level.

`PeskAAS` was designed to be a free, online data exploration tool to enable rapid development of simple, near-real time fisheries monitoring systems for accurate and timely fish production data to use in adaptive, evidence-based management of fisheries resources. The tracking hardware and data service from Pelagic Data Systems (PDS) is an optional extra and represents additional costs to governments that must be considered in terms of sustainability. All the analytics of `PeskAAS` relating to CPUE are done without PDS data. We found effort

estimations that fishers provided if asked were sufficiently accurate, and a VAC can be calculated from this by averaging all trip durations recorded from certain fishers. The specific values of the vessel tracking functionality were 1) an understanding of effort and activity distribution and, 2) the ability to calibrate effort through dynamic VAC calculation over time, and 3) in certain contexts (not tested here) may provide a safety at sea function.

If collected and managed in inclusive processes with fishers, fishing location data can be used by fishers to better argue and prove tenure and/or use rights (i.e., in line with the FAO Voluntary Guidelines on fisheries [28] and tenure [29]), for example in marine spatial planning processes that can otherwise ignore complex socio-political inequalities [30]. This value was realised in the case of the TS Taipei container ship grounding in Taiwan, where spatial and catch records from small-scale fishers were utilised to leverage fair reparations to fishers from the shipping company [14]. To date the spatial information obtained from combined vessel tracking with PeskAAS has not been used for coastal zone management, but protecting the rights of Timorese fishers to marine areas is a key commitment in the new draft National Fisheries Strategy (2018) that the PeskAAS data and team helped to inform. In other contexts, there are concerns that such data might work against fisher livelihoods, i.e. in nations and regions where exclusive no take zones and top down compliance are preferred strategies [31, 32]. The intended and unintended implications of data innovations should be considered from the outset of any such pilot and subsequent scaling out phases.

The co-design of PeskAAS with fisheries managers was crucial in building a financially and institutionally sustainable system. This same user-centred design process ensured that the PeskAAS analytical workflow was not initially too complex, and only included statistics and insights that could be readily interpreted by fisheries management users. Hence, in scaling this tool to other countries and contexts, the need to build local legitimacy and a feeling of ownership amongst government stakeholders will necessitate a process of co-design. The technical capacity in Timor-Leste for digital systems is improving, but as in most low-income country governments, the long-term maintenance of the system will still require external support. Ongoing work on PeskAAS will develop its modularity and customisation through templates for catch forms, analytics package integrations (such as TropFishR [8]) and dashboard tutorials to guide managers, to allow for more flexibility to other fish production systems, countries and capacities.

PeskAAS was developed as an open source script to make it highly adaptable to different fisheries contexts and strategic objectives. The catch form and scripted analytics require no specialised equipment and the dashboard is displayed in a web browser. However, we acknowledge that the scripted nature of PeskAAS limits its use to those with programming expertise, or implies a need to hire help that may be uncommon or unavailable in low-income countries. Rather than be an off-the-shelf solution, we envision PeskAAS to be a software framework that external NGOs or multilateral agencies can utilise to establish a highly contextualised, user-centred system for obtaining and using data from small-scale fisheries to guide decision-making. Data generated by PeskAAS were used to show how fishing technology could improve fisher incomes and fish availability in Timor-Leste [27], which substantiates government investment in interventions that are more cognisant and sensitive to SSF. Future work will focus on developing modules and templates that can be populated by local actors in new fisheries contexts to reduce the coding workload and streamline implementation, but we anticipate some level of technical assistance will continue to be required.

PeskAAS in Timor-Leste is open access and publicly available. In theory, anyone with access to a computer, tablet or smartphone, and an internet connection, are able to see the data aggregated to the municipal level. The reality is that in Timor-Leste, as in many low-income countries, there are still substantial barriers (e.g., cost of technology, digital literacy, formal identification) to digital inclusion (the access to and use of ICTs) that limit the extent

to which fishers can access and benefit from such data. Addressing these limitations represents challenges for future pilots where participatory action research principles would be relevant [33] and where initiative objectives include empowering individual fishers in management decision-making, or improving their livelihoods through fisher-level business planning. This was beyond the scope of the pilot we describe here.

Further capacity building is also necessary to ensure `PeskAAS` outputs are being correctly interpreted to inform policy and guide extension services delivered by government actors to and for fishing communities in Timor-Leste. Better data systems can only drive informed decision-making if the information is available and understandable. Currently, this PeskAAS is suited to boat-based fisheries, compounding the unawareness and exclusion of foot fisheries often conducted by women [34]. To get a full picture of SSF and enable local legitimacy and sustainability, Timor-Leste must build on initial successes with co-management [35] as a platform on which to base a range of data collection methods and strategies. This will ensure stakeholders are included in contributing local ecological knowledge and forming shared understanding and strategies for stewardship of their aquatic resources.

In conclusion, the pilot of `PeskAAS` enabled system refinements, illuminated limitations and developed preliminary data for decision makers. The data have been taken into consideration in national-level fisheries governance, through the new draft National Fisheries Strategy (2018) and revised Fisheries Decree Law (2019). The application has potential for government and non-government managers, researchers and students looking for a low cost solution to address SSF data gaps.

## Supporting information

**S1 Data. List of species, species groups and family categories used for recording catch along with a and b parameters used to calculate weight from length measurements.** (DOCX)

**S2 Data. MySQL relational database schema with cascading foreign keys shown by dashed arrows.** Figure generated in MySQL Workbench (Oracle Corp.). (DOCX)

**S3 Data. PeskaPARSE.R' script file.** The PeskaPARS.R script scheduled on a daily cron job, run on a Google Cloud Platform virtual machine. (R)

## Acknowledgments

We are very grateful to the Timorese fishers and community members who collaborated with us and Timor-Leste's Ministry of Agriculture and Fisheries in this project. Particular thanks go to: Joctan Dos Reis Lopes for leading the `PeskAAS` team on the ground; to the WorldFish Timor-Leste team members Dave Mills, Mario Gomes, Mario Pereira and Agustinha Duarte for providing important local insights throughout the co-design process; to Pedro Rodrigues, Lucas Fernandes and the Fisheries Directorate staff for helping us to build from the ground up; and to Pip Cohen for insight and comments on this manuscript.

## Author Contributions

**Conceptualization:** Alexander Tilley.

**Data curation:** Alexander Tilley, Joctan Dos Reis Lopes, Shaun P. Wilkinson.

**Formal analysis:** Alexander Tilley, Joctan Dos Reis Lopes.

**Funding acquisition:** Alexander Tilley.

**Methodology:** Alexander Tilley, Joctan Dos Reis Lopes, Shaun P. Wilkinson.

**Project administration:** Alexander Tilley.

**Software:** Shaun P. Wilkinson.

**Supervision:** Alexander Tilley.

**Validation:** Joctan Dos Reis Lopes.

**Visualization:** Alexander Tilley, Shaun P. Wilkinson.

**Writing – original draft:** Alexander Tilley, Shaun P. Wilkinson.

**Writing – review & editing:** Alexander Tilley, Shaun P. Wilkinson.

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
