## [Decision Letter · Decision Letter 0]

15 Jul 2020

PONE-D-20-16520

PeskAAS: A near-real-time, open-source monitoring and analytics system for small-scale fisheries

PLOS ONE

Dear Dr. Tilley,

Thank you for submitting your manuscript to PLOS ONE. After careful consideration, we feel that it has merit but does not fully meet PLOS ONE’s publication criteria as it currently stands. Therefore, we invite you to submit a revised version of the manuscript that addresses the points raised during the review process.

ACADEMIC EDITOR: I read this MS with interest as it addresses an important gap in small-scale fisheries which require precision and reliability to address sustainable management of the fisheries. I am sure that the reviews it has received will greatly improve its quality. And, so, please address all the queries, especially of data flow as indicated by Reviewer #2.

We look forward to receiving your revised manuscript.

Kind regards,

Ismael Aaron Kimirei, Ph.D.

Academic Editor

PLOS ONE

Journal Requirements:

2. We note that [Figure(s) 3 and 4] in your submission contain [map/satellite] images which may be copyrighted. All PLOS content is published under the Creative Commons Attribution License (CC BY 4.0), which means that the manuscript, images, and Supporting Information files will be freely available online, and any third party is permitted to access, download, copy, distribute, and use these materials in any way, even commercially, with proper attribution. For these reasons, we cannot publish previously copyrighted maps or satellite images created using proprietary data, such as Google software (Google Maps, Street View, and Earth). For more information, see our copyright guidelines: http://journals.plos.org/plosone/s/licenses-and-copyright.

1.    You may seek permission from the original copyright holder of Figure(s) [3 and 4] to publish the content specifically under the CC BY 4.0 license. 

3. In your Methods section, please provide additional location information of the study sites, including geographic coordinates for the data set if available.

4. In your Methods section, please provide additional information regarding the permits you obtained for the work. Please ensure you have included the full name of the authority that approved the study sites access and, if no permits were required, a brief statement explaining why.

5. Please provide additional details regarding participant consent. In the ethics statement in the Methods and online submission information, please ensure that you have specified (1) whether consent was informed and (2) what type you obtained (for instance, written or verbal, and if verbal, how it was documented and witnessed).

6.Thank you for stating the following in the Competing Interests section:

[The authors have declared that no competing interests exist.].   

We note that one or more of the authors are employed by a commercial company: Wilderlab, Wellington

Reviewers' comments:

Reviewer's Responses to Questions

**Comments to the Author**

1. Is the manuscript technically sound, and do the data support the conclusions?

Reviewer #1: Yes

Reviewer #2: Yes

Reviewer #3: Yes

2. Has the statistical analysis been performed appropriately and rigorously? 

Reviewer #1: Yes

Reviewer #2: Yes

Reviewer #3: Yes

3. Have the authors made all data underlying the findings in their manuscript fully available?

Reviewer #1: Yes

Reviewer #2: Yes

Reviewer #3: Yes

4. Is the manuscript presented in an intelligible fashion and written in standard English?

Reviewer #1: Yes

Reviewer #2: Yes

Reviewer #3: Yes

5. Review Comments to the Author

Reviewer #1: Review notes:

The manuscript introduces a new tool for small-scale fisheries data collation and analysis using an Rshiny platform. The manuscript is well written and the introduction does a good job of describing the many issues with monitoring small-scale fisheries. The tool itself is sophisticated and is now being used by the Timor-Leste government in monitoring their fisheries. This is a fantastic example of an operational fisheries tool and will make a great addition to the literature.

The tool appears to primarily be used in monitoring catch and effort, rather than providing analytical assessments for fisheries managers (i.e. stock assessment). This is of course fine (and much needed in the case of small scale fisheries), but I think perhaps this specific scope of the tool needs to be stated clearly somewhere, perhaps in the abstract. Or perhaps a paragraph in the discussion could envision the next steps for this tool/data in fisheries management? I.e. what comes next after appropriate monitoring?

Much of the manuscript focused on describing the tool pipeline, from shore-based observers to the final Rshiny interface. The description of this pipeline was logical and detailed, but I did have a few line-specific comments below that request more information in some sections to help improve potential reproducibility. Particularly, for Ln 171 – 184 this seems to be the biggest analytical component of the tool but the details and results are limited (see broader comment below).

Finally, the introduction makes the case for a ‘light touch’ tool for data collection and analytics. While I agree with their logic in the introduction, I don’t quite agree that peskAAS is ‘light touch’. For example, the peskAAS application in Timor-Leste involved observers with tablet computers, gps trackers on boats, sophisticated coding in R, plus multiple software subscription services (e.g. Rshiny and Kobo toolbox). The authors do detail some of the limitations in the discussion, but I think more could be done to address the scalability and ‘light touch’ effectiveness of this tool.

Abstract

- Would be useful to indicate what type of data is needed to operate peskAAS?

- Needs to explain that the peskAAS was developed and implemented for small scale fisheries in Timor-Leste

- Growth parameters doesn’t seem like the correct term to me (see comment below). Suggest using ‘length-weight’.

Introduction

- Ln 41: can you list some other tools that are used in management? E.g. fishpath (https://www.fishpath.org/the-tool), stock synthesis etc.

- Ln 41: there is no closing parenthesis

- Reference 15 – title is incorrect here. Also, is this peer-reviewed? If not then I suggest not using it as a reference

- Ln 52: need a closing parenthesis

-

peskAAS

- Ln 76: more information needed on spline type here

- Ln 89: is this the figure caption? More text is needed here to better explain the pipeline, and all the acronyms and terms. E.g. PDS, DB, API,

- Ln 91: list tablet model

- Ln 92: can you add a sentence describing what Kobotool box is?

- Ln 97: what are the gear type options?

- Ln 99: what size was measured (fork length, standard length, total length) and what units?

- Ln 99: units for trip duration?

- Ln 114: I couldn’t find this supp file 2

- Ln 118: I’m not sure if this is an error, or perhaps I don’t have the right supp file. But the VBGF is typically used to get size-at-age, using parameters k, t, and l-infinite. The parameters in table supplementary information 2 are for a and b, and are described as being used to convert length-to-weight. Suggest not using the VBGF term but rather simply saying ‘length-to-weight’ conversion.

- Ln 124: does the italics of ‘via’ have special meaning here? It’s not clear what it is

- Ln 135: I didn’t know what

- Ln 143-144: roughly how many trips to the shore-based data collectors sample?

- Ln 145: insert model number of the gps units (as per typical scientific notation)

- Ln 146: was it only voluntary boats that had gps installed? Or is this mandated at a local/state/national level?

- Ln 147-148: great figure. Is the concentration of effort near ports a function of effort or the range of cellular towers?

- Ln 157: wouldn’t it make more sense to divide by the number of fishers?

- Ln 167: isn’t cpue kg per hour per fisher? Or is the per fisher component handled in the EPT? Some clarification needed here.

- Equation 1: why multiply by 0.001?

- Ln 171 – 184: this seems to be the biggest analytical component of the tool but the details and results are sparse. Suggest adding more methods details and a results figure/table of the cross-validation test. Specifically: (1) clarify if this analysis is done as part of the peskAAS pipeline, or if this is a secondary analysis for validation; (2) can you add references to justify that this was an appropriate method to fill in missing attributes?; (3) what temporal spacing did you use? (ln 177); (4) was the 80%/20% cross validation only performed once? If so this is not adequate. Typically, this random split is repeated multiple times (e.g. 10 times) and the performance validated each time. Or you could use k-folds validation.

Ln 198-200: this is really great to see it is operational.

Reviewer #2: General comments:

This manuscript nicely lays out the development of an online or desktop tool to visualize fisheries-dependent data for use in management and decision making. This tool is already being incorporated into fisheries management in Timor-Leste, indicating its utility and need for supporting small scale fisheries. This manuscript is well done but I would encourage the authors to consider a couple aspects to help highlight where this tool could be useful and to clarify how it functions.

1. Framing: I would encourage the authors to consider how they contextualize where this tool would be useful. The abstract highlights the near real-time production data being aggregated through this tool, indicating an ability to use it for adaptative management. However, the introduction highlights its utility in gaining an understanding of food-security (based on production) while the discussion highlights utility in marine spatial planning. This tool could help in all three of these contexts and they rightly deserve to be highlighted. However, I would encourage the authors to provide more connection on these points being made between sections.

For instance: fish as a base for food security requires effective management for lasting security. Small scale fisheries afford the opportunity to provide food security but are difficult to manage because of a lack of data collection and dispersed nature of the fishery. And in order to sustain food security through development it is necessary to understand where fishing effort predominately takes place.

2. Data collection and flow: I had trouble understanding the data flow for this tool. It is not very clear if the spatial and temporal aspects of the data are because of the VMS data or if the shore side sampling provides a spatial component depending on which port the data were collected, it seems only the VMS data provide the spatial component. The VMS data appear to be a critical aspect of the functioning and ultimate use of this tool. However, this feature is called an “optional extra” on line 204. The authors do point out how this would change the function of the tool, however, the abstract and discussion highlight the data gaps around spatial fisheries data. Therefore, it seems that the VMS component is critical for the use of this tool. I understand it will function without the VMS data, so it is important to understand all the data incorporated and how it fits the potential uses outlined in point 1 above.

I would note that the description of data collection and flow is highlighted in the Figure 2 caption (lines 90 – 101 after the references). This information should be in the main text as well. Additionally, Figure 2 doesn’t appear to be referenced in the manuscript.

Specific Comments

Line 27: Missing citation for “small scale fisheries are responsible for half of the global fish catch”

Line 69: Where do the landings records come from?

Lines 73-74. The resolution for Figure 1 is too low to be able to read the screen shot of what the app can do. As a side note about the app: upon taking a look at the shinyapp, I wonder if it would be more intuitive to have none of the satellite boxes checked upon opening the app and allowing the user to specify which data they want to see instead of having to deselect everything and then select.

Lines 75-78: There is not enough detail about the smoothing spline or how a user should decide whether to use this function. Equation 1 at line 163 does not include the smoothing spline either, therefore it is not possible to decipher how this aspect of the tool functions.

Line 114: It looks like supplementary information 2 is a species list with growth parameters, am I missing supplementary file 2 to support ‘peskaPARSE.R’?

Line 143: Are shore-based data collectors how catch data are normally collected for these fisheries? The information provided under the Figure 2 Caption titled Catch document at landing sites is very helpful context. I would recommend this being added to the methods to help readers understand data collection a little more clearly. I also am not seeing Figure 2 (or Figure 3) referenced in the text, I see Figure 1 at line 73 and then Figure 4 at line 148.

Line 146: were the GPS/VMS units that were placed on the 5-15 boats per landing site placed on both power and canoe vessels?

L

ine 147: Is it costly to the harvesters to upload VMS data via cell phone? For people interested in implementing this system it could be helpful to get an idea of the overall cost to harvesters. Is there also a logbook on the cellphone that harvesters can record catch that then gets associated with the location data or that only comes from the shore side data?

Line 154: what does relative fishing success mean? I would recommend rephrasing this for clarity.

Line 188: are there a minimum number of observations that are needed to be able to use the data? For instance, US federal fisheries require a minimum of 3 observations to be able to aggregate data otherwise it can’t be shown. This would mean some data are left out, is this the case for Timor-Leste? It would be good to know if not all data are shown.

Lines 198 – 200: I personally would like to have a bit more detail about how this tool is being utilized by the Timorese government than what is provided.

Line 204: the spatial data from VMS seems critical, unless the port samples are also considered to be part of the spatial data (doesn’t seem like it). Are there area closures or other types of management that make this spatial data critical to adaptive management.

Lines 205-207 indicate the fishery footprint data are most useful for coastal zone planning, are they not used in the

current fisheries management practices?

Figures 2 and 3 are not cited in the text from what I can tell.

It is not clear what Supplementary Information 2 is supporting

Reviewer #3: Dear Authors,

Thank you very much for an interesting paper describing this new system for an open-source monitoring and analytics system. I agree that especially for small-scale fisheries it is a great problem to have reliable landings and catch data as a very important first step to establish some kind of fisheries management.

I think your system well developed and you present the results in an appropriate fashion supported by the data. The results show that this could be a very useful tool for collecting data for fisheries managers. I have, however, some doubts how this system could be established in practice in low income countries. For the development of your system and to show how it works you have selected Timor-Leste as an example. For this trial you have organised the data collection by hiring data collectors (at least that how it is described) and I am asking myself how this should work in the long run. Should the fishers report the data themselves (what I assume from your descriptions)? What will be the incentives of the fishers to participate in such a system? Misreporting is a problem in all fisheries management systems around the world.

Therefore, could you describe a bit better how you foresee that this system will be used afterwards, who need to participate in the system and what the authorities can do with this information. As I am not a fisheries biologist, I am also not sure about the fishbase database. Is the catch weight (average of all fishes in the catch, the single fish weights?) really the only parameter you need to judge the conditions of the fish stocks in mixed fisheries?

I have a few additional more detailed comments regarding your paper:

Line 36 ff.: I agree that it is a problem in low income countries to collect data on catch/landings but the question is if those data alone would help to overcome the malnutrition in many countries due to the ability to establish a management system. I think we are still far away from introducing a fisheries management system with catch and/or landings data alone. Especially for mixed fisheries I believe that another approach could work. Many coastal communities have established some kind of management strategies as they have usually a substantial knowledge about the resources and have established some kind of self-management (e.g. in a community-based-management system). Is there information on self-management in the fishing communities in Timor-Leste?

Lines 154 ff.: How does e.g. the number of fishers on a vessel influence the input data (fisher-hours)? The CPUE is very much depending on vessels size, fishing gear employed, number of fishers on board, etc. In small-scale fisheries often the owner alone is fishing or only a small number of fishers fish together.

Lines 166 ff.: I cannot judge how good those estimated numbers are. My guess is, however, that there are so many factors influencing catch levels that this may squeeze everything too much into a certain routine calculation. However, I am aware that this may be the best methodology available in this case.

Lines 183-185: Looks good but still I have doubts about the overall effort necessary to get all this detailed information and if there are no other possibilities.

6. PLOS authors have the option to publish the peer review history of their article (what does this mean?). If published, this will include your full peer review and any attached files.

Reviewer #1: No

Reviewer #2: No

Reviewer #3: No

---

## [Author Response · Author response to Decision Letter 0]

19 Sep 2020

(This has also been provided in the form of the "Response to reviewers" file)

Thank you very much for taking the time to carefully review our manuscript. Your comments have been extremely helpful in improving and clarifying the paper. Please find our responses to each of your comments below (in blue) with further clarification where appropriate, and the line numbers of any subsequent changes made to the ms.

PONE-D-20-16520

PeskAAS: A near-real-time, open-source monitoring and analytics system for small-scale fisheries

ACADEMIC EDITOR: I read this MS with interest as it addresses an important gap in small-scale fisheries which require precision and reliability to address sustainable management of the fisheries. I am sure that the reviews it has received will greatly improve its quality. And, so, please address all the queries, especially of data flow as indicated by Reviewer #2.

Thank you. We have addressed all of the reviewers’ helpful comments, stating below 

2. We note that [Figure(s) 3 and 4] in your submission contain [map/satellite] images which may be copyrighted. We require you to either (1) present written permission from the copyright holder to publish these figures specifically under the CC BY 4.0 license, or (2) remove the figures from your submission:

Fig 3 was drawn as a vector image by AT so does not require copyright information. Copyright information has been added to the caption of figure 4. 

3. In your Methods section, please provide additional location information of the study sites, including geographic coordinates for the data set if available.

Given the national scale of the study, Figure 3 has been amended to include lat/long grid coordinates instead of providing coordinates for each of the landing sites.

4. In your Methods section, please provide additional information regarding the permits you obtained for the work. Please ensure you have included the full name of the authority that approved the study sites access and, if no permits were required, a brief statement explaining why.

5. Please provide additional details regarding participant consent. In the ethics statement in the Methods and online submission information, please ensure that you have specified (1) whether consent was informed and (2) what type you obtained (for instance, written or verbal, and if verbal, how it was documented and witnessed).

We have added the following statement from line 137. ”All research was authorised and conducted in partnership with the Timor-Leste Ministry of Agriculture and Fisheries. All catch data collection was carried out with the verbal consent of the fisher. All participation in catch data monitoring and vessel tracking was voluntary, and fishers not willing to participate were not prejudiced against in any way.”

6.Competing Interests section:

We note that one or more of the authors are employed by a commercial company: Wilderlab, Wellington

Please also include the following statement within your amended Funding Statement. “The funder provided support in the form of salaries for authors [insert relevant initials], but did not have any additional role in the study design, data collection and analysis, decision to publish, or preparation of the manuscript. The specific roles of these authors are articulated in the ‘author contributions’ section.” If your commercial affiliation did play a role in your study, please state and explain this role within your updated Funding Statement.

Funding statement updated as required from line 326. This has been included in the cover letter along with the updated competing interests statement.

2. Please also provide an updated Competing Interests Statement declaring this commercial affiliation along with any other relevant declarations relating to employment, consultancy, patents, products in development, or marketed products, etc. Within your Competing Interests Statement, please confirm that this commercial affiliation does not alter your adherence to all PLOS ONE policies on sharing data and materials by including the following statement: "This does not alter our adherence to PLOS ONE policies on sharing data and materials.”.

Declared as required

We have amended the in-text citation and caption of supplementary information 3 and provided the associated Supplementary information 3 file (PeskaParse.R script) that was accidentally omitted when the paper was initially submitted. 

Reviewer #1: Review notes:

The manuscript introduces a new tool for small-scale fisheries data collation and analysis using an Rshiny platform. The manuscript is well written and the introduction does a good job of describing the many issues with monitoring small-scale fisheries. The tool itself is sophisticated and is now being used by the Timor-Leste government in monitoring their fisheries. This is a fantastic example of an operational fisheries tool and will make a great addition to the literature.

The tool appears to primarily be used in monitoring catch and effort, rather than providing analytical assessments for fisheries managers (i.e. stock assessment). This is of course fine (and much needed in the case of small scale fisheries), but I think perhaps this specific scope of the tool needs to be stated clearly somewhere, perhaps in the abstract. Or perhaps a paragraph in the discussion could envision the next steps for this tool/data in fisheries management? I.e. what comes next after appropriate monitoring?

Thank you. Yes, complicated stock assessment analytics were not feasible for the capacity level of fisheries managers in Timor when PeskAAS was developed. However, we are currently developing new integrations of statistical tools and techniques into the analytical workflow. We have made this clearer in the abstract (lines 18-20) and throughout the discussion (lines 276-79).

Much of the manuscript focused on describing the tool pipeline, from shore-based observers to the final R shiny interface. The description of this pipeline was logical and detailed, but I did have a few line-specific comments below that request more information in some sections to help improve potential reproducibility. Particularly, for Ln 171 – 184 this seems to be the biggest analytical component of the tool but the details and results are limited (see broader comment below).

This has been clarified in the ms (lines 219-33) and all of the code of the nearest neighbor analysis is provided so can be interpreted and replicated when necessary. 

Finally, the introduction makes the case for a ‘light touch’ tool for data collection and analytics. While I agree with their logic in the introduction, I don’t quite agree that peskAAS is ‘light touch’. For example, the peskAAS application in Timor-Leste involved observers with tablet computers, gps trackers on boats, sophisticated coding in R, plus multiple software subscription services (e.g. R shiny and Kobo toolbox). The authors do detail some of the limitations in the discussion, but I think more could be done to address the scalability and ‘light touch’ effectiveness of this tool.

This is certainly true and we concur that it has not been a light touch engagement to this point four years later. We have added a more thorough discussion of limitations and strengths of the tool for scaling in lines 280-290.

Abstract

- Would be useful to indicate what type of data is needed to operate peskAAS?

- Needs to explain that the peskAAS was developed and implemented for small scale fisheries in Timor-Leste

Amendments have been made to the abstract to clarify these two points (lines 13-20)

- Growth parameters doesn’t seem like the correct term to me (see comment below). Suggest using ‘length-weight’.

Changed as suggested

Introduction

- Ln 41: can you list some other tools that are used in management? E.g. fishpath (https://www.fishpath.org/the-tool), stock synthesis etc.

Added reference to specific tools (line 52)

- Ln 41: there is no closing parenthesis

- Reference 15 – title is incorrect here. Also, is this peer-reviewed? If not then I suggest not using it as a reference

This is a Master's thesis from the University of Malmo so not peer-reviewed, but we feel suitable for citation. The reference has been updated to ensure it is complete. 

- Ln 52: need a closing parenthesis

Done

- Ln 76: more information needed on spline type here

We expanded the sentence to: “cubic smoothing splines weighted by trip effort to the CPUE data with a user defined smoothing parameter to aid in visualization”. Lines 91-2

- Ln 89: is this the figure caption? More text is needed here to better explain the pipeline, and all the acronyms and terms. E.g. PDS, DB, API,

Caption expanded with more details and acronyms defined

- Ln 91: list tablet model

Data can be collected on any tablet or phone running android. We used many different types so we have added that these were android tablets. We have made this clearer in the text (line 117)

- Ln 92: can you add a sentence describing what Kobotool box is?

Done

- Ln 97: what are the gear type options?

Added

- Ln 99: what size was measured (fork length, standard length, total length) and what units?

Added

- Ln 99: units for trip duration?

Added

- Ln 114: I couldn’t find this supp file 2

This should have been labelled “Supplementary information 3”, and the file was accidentally omitted. We have provided the PeskaPARSE.R script with the revision submission as Supplementary information 3.

- Ln 118: I’m not sure if this is an error, or perhaps I don’t have the right supp file. But the VBGF is typically used to get size-at-age, using parameters k, t, and l-infinite. The parameters in table supplementary information 2 are for a and b, and are described as being used to convert length-to-weight. Suggest not using the VBGF term but rather simply saying ‘length-to-weight’ conversion.

Changed as suggested

- Ln 124: does the italics of ‘via’ have special meaning here? It’s not clear what it is

Changed to non italics

- Ln 135: I didn’t know what

Comment incomplete?

- Ln 143-144: roughly how many trips to the shore-based data collectors sample?

This is stated in line 239 (= 29,251 trips up to July 2020). The average number of trips sampled per day across all enumerators is 3. 

- Ln 145: insert model number of the gps units (as per typical scientific notation)

Entered

- Ln 146: was it only voluntary boats that had gps installed? Or is this mandated at a local/state/national level?

Tracking was voluntary. This has been clarified in the ms in lines 139 and 187

- Ln 147-148: great figure. Is the concentration of effort near ports a function of effort or the range of cellular towers?

No, this has nothing to do with cell range because the trackers are archival, meaning that when they are not in range of a cell tower they store location data on the device (for up to one year). The following text has been added to the caption to clarify: “Figure 4. The relative effort heat map of small-scale fishing vessel tracks between August 2018 and February 2020 in Timor-Leste from the PeskAAS dashboard. Red shading represents areas of highest effort (near to homeports and shore), and orange to yellow shading shows areas fished with decreasing effort further from shore. Basemap obtained from Esri ArcGIS via leaflet [23].” 

- Ln 157: wouldn’t it make more sense to divide by the number of fishers?

Instead of multiplying? No, because if you have boat ‘A’ with 1 fisher fishing for 1 hour compared to boat ‘B’ with 2 fishers fishing for 1 hour, boat B will be fishing the same time but with double the effort. If we divided 1 hour by 2 fishers this would be 0.5, so half the effort of boat A, which would be incorrect.

- Ln 167: isn’t cpue kg per hour per fisher? Or is the per fisher component handled in the EPT? Some clarification needed here.

Clarified to show kg per fisher-hour

- Equation 1: why multiply by 0.001? 

To return tonnes from kg

- Ln 171 – 184: this seems to be the biggest analytical component of the tool but the details and results are sparse. Suggest adding more methods details and a results figure/table of the cross-validation test. Specifically: (1) clarify if this analysis is done as part of the peskAAS pipeline, or if this is a secondary analysis for validation; (2) can you add references to justify that this was an appropriate method to fill in missing attributes?; (3) what temporal spacing did you use? (ln 177); (4) was the 80%/20% cross validation only performed once? If so this is not adequate. Typically, this random split is repeated multiple times (e.g. 10 times) and the performance validated each time. Or you could use k-folds validation.

Thank you for pointing out these shortcomings. We have changed the tense in the section to reflect that the nearest neighbor assignment occurs as part of the pipeline, and that we validated it on the data available up until 31 July 2020 (the code for the initial validation test is included under comment in the app.R file and the code for ongoing assignment of gear and habitat types is included in the peskaPARSE.R file (Supplementary file 3)). 

The temporal spacing was distributed at even intervals over the course of the trip to allow for both spatial and temporal similarities in trip comparisons (Ms text has been updated to: ‘To achieve this, ten evenly temporally-spaced, two-dimensional GPS vectors from each trip are used to generate training- and query-data matrices’). We re-ran the analysis to iterate 100x rather than once, and we present both mean and stdev prediction success rates (thanks for pointing this out). We tried a few different classification methods and settled on NN based on predictive success and simplicity - (Cover & Hart 1967) in the text. (Lines 213-33)

Ln 198-200: this is really great to see it is operational.

Thank you

Reviewer #2: General comments:

This manuscript nicely lays out the development of an online or desktop tool to visualize fisheries-dependent data for use in management and decision-making. This tool is already being incorporated into fisheries management in Timor-Leste, indicating its utility and need for supporting small-scale fisheries. This manuscript is well done but I would encourage the authors to consider a couple aspects to help highlight where this tool could be useful and to clarify how it functions.

1. Framing: I would encourage the authors to consider how they contextualize where this tool would be useful. The abstract highlights the near real-time production data being aggregated through this tool, indicating an ability to use it for adaptive management. However, the introduction highlights its utility in gaining an understanding of food-security (based on production) while the discussion highlights utility in marine spatial planning. This tool could help in all three of these contexts and they rightly deserve to be highlighted. However, I would encourage the authors to provide more connection on these points being made between sections.

For instance: fish as a base for food security requires effective management for lasting security. Small scale fisheries afford the opportunity to provide food security but are difficult to manage because of a lack of data collection and dispersed nature of the fishery. And in order to sustain food security through development it is necessary to understand where fishing effort predominately takes place.

Thank you for this insight. We have improved consistency between sections and brought this together better throughout the discussion (lines 256-292) 

2. Data collection and flow: I had trouble understanding the data flow for this tool. It is not very clear if the spatial and temporal aspects of the data are because of the VMS data or if the shore side sampling provides a spatial component depending on which port the data were collected, it seems only the VMS data provide the spatial component. 

The VMS data appear to be a critical aspect of the functioning and ultimate use of this tool. However, this feature is called an “optional extra” on line 204. The authors do point out how this would change the function of the tool, however, the abstract and discussion highlight the data gaps around spatial fisheries data. Therefore, it seems that the VMS component is critical for the use of this tool. I understand it will function without the VMS data, so it is important to understand all the data incorporated and how it fits the potential uses outlined in point 1 above. 

There are two levels of spatial components to the system: 

1) Municipal: Enumerators are operating in different municipalities of the country, so the catch and effort can be summarised and grouped at this level to evaluate variations in fishing patterns, gears, species, habitats etc. by area of the country. For example, the north and south coasts of the country differ significantly in terms of species assemblage, habitat, bathymetry etc. 

2) The movement tracks from the VMS devices provide very high resolution data of the activity space of each individual fishing vessel and hence the overall spatial extent of the national fishery. PDS tracking also enables the accurate measurement of effort in trip length (hours). However, at this point there is no integrated mechanism to detect at what time during the trip the gear was hauled or the catch was landed. Catch is only recorded when it reaches the shore and recorded by an enumerator. 

The addition of VMS is not a critical aspect of the system. It works completely without vessel tracking, however, the ability to visualise fishing on a map is beneficial and popular. We have added clarification of exactly what value the PDS tracking adds to the PeskAAS system in the discussion in lines 248-268.

I would note that the description of data collection and flow is highlighted in the Figure 2 caption (lines 90 – 101 after the references). This information should be in the main text as well. Additionally, Figure 2 doesn’t appear to be referenced in the manuscript.

I think there was an error with the formatting of the original file because this section “Catch documentation at landings sites” was intended as a section in the methods, and is not a figure caption. We have added “METHODS” at the start of the section to clarify the paper structure. Figure 3 is referenced at the start of this paragraph (originally line 95). Figure 2 is referenced just before the figure itself (originally line 87).

Specific Comments

Line 27: Missing citation for “small scale fisheries are responsible for half of the global fish catch”

Inserted

Line 69: Where do the landings records come from?

Clarified in extended figure 2 caption and in lines 82-83 and 115-135

Lines 73-74. The resolution for Figure 1 is too low to be able to read the screenshot of what the app can do. As a side note about the app: upon taking a look at the shinyapp, I wonder if it would be more intuitive to have none of the satellite boxes checked upon opening the app and allowing the user to specify which data they want to see instead of having to deselect everything and then select.

We have replaced the screenshot with a higher-resolution version, but the website is live and open access so the best way of exploring functionality would be to visit the website itself (the URL is provided in the caption). We did start out having none of the tick boxes checked but changed to checking all by default after consulting with the primary user (Timor-Leste Ministry of Fisheries).

Lines 75-78: There is not enough detail about the smoothing spline or how a user should decide whether to use this function. Equation 1 at line 163 does not include the smoothing spline either, therefore it is not possible to decipher how this aspect of the tool functions.

We added the smoothing spline function just to help users better visualize the data - rather than to generate statistically robust trends/time series. We are hesitant to provide guidance on setting the smoothing parameter - since this will depend a lot on the application and what insight the user is looking for.

We have added additional details in the text about how the smoothing spline is calculated: “...a slider widget is included for users to fit cubic smoothing splines weighted by trip effort to the CPUE data with a user defined smoothing parameter that is dynamically adjustable to aid in visualization” lines 90-94

Line 114: It looks like supplementary information 2 is a species list with growth parameters, am I missing supplementary file 2 to support ‘peskaPARSE.R’?

Our apologies. This should have been labelled Supplementary file 3, and the file was accidentally omitted in any case. We have provided the PeskaPARSE.R script with the revision submission as Supplementary file 3.

Line 143: Are shore-based data collectors how catch data are normally collected for these fisheries? The information provided under the Figure 2 Caption titled Catch document at landing sites is very helpful context. I would recommend this being added to the methods to help readers understand data collection a little more clearly. I also am not seeing Figure 2 (or Figure 3) referenced in the text, I see Figure 1 at line 73 and then Figure 4 at line 148.

I think there must have been an error with the original file as this section “Catch documentation at landings sites” is a section in the methods already, and is not a figure caption. We have added “METHODS” at the start of the section to clarify the paper structure. Figure 3 is referenced at the start of this paragraph (originally line 95). Figure 2 is referenced just before the figure itself (originally line 87).

Line 146: were the GPS/VMS units that were placed on the 5-15 boats per landing site placed on both power and canoe vessels?

Yes to both. Clarified in text.

Line 147: Is it costly to the harvesters to upload VMS data via cell phone? For people interested in implementing this system it could be helpful to get an idea of the overall cost to harvesters. Is there also a logbook on the cellphone that harvesters can record catches that then gets associated with the location data or that only comes from the shore side data?

There is zero cost to the fishers (line 187) Currently no data are recorded by the fishers themselves, but they could if trained to use the app. All catch data are recorded by government enumerators at landing sites on shore, and all geolocation data are recorded and communicated automatically by the VMS unit without any input from the fisher. Currently in Timor-Leste fisher participatory monitoring is only possible when smartphones can be provided, and even then literacy and capacity levels inhibit the scale of the work. However, an app that fishers can collect, hold and use their own data to improve efficiency and reduce costs is planned in the coming years. This is discussed in lines 280-292

Line 154: what does relative fishing success mean? I would recommend rephrasing this for clarity. 

Rephrased to “To quantify how catch volume varied over time and space, the catch-per-unit-effort (CPUE) is used as a metric for relative fish abundance, and to track the response of fish stocks to fishing pressure [24]” (lines 200-1)

Line 188: are there a minimum number of observations that are needed to be able to use the data? For instance, US federal fisheries require a minimum of 3 observations to be able to aggregate data otherwise it can’t be shown. This would mean some data are left out, is this the case for Timor-Leste? It would be good to know if not all data are shown.

We consulted with the Timorese government on the issue of privacy and data granularity, and it was agreed that we would not reveal fisher’s details but that individual landing and track records were acceptable to present on the application. Hence, all data are present in the app regardless of minimum observation number. 

Lines 198 – 200: I personally would like to have a bit more detail about how this tool is being utilized by the Timorese government than what is provided.

We have added text to the discussion to detail current and recent uses of the application and data generated (lines 240-3, 261-5, 284-8, and 312-4). 

Line 204: the spatial data from VMS seems critical, unless the port samples are also considered to be part of the spatial data (doesn’t seem like it). Are there area closures or other types of management that make this spatial data critical to adaptive management?

There are some marine protected areas, but management is very limited and currently PDS tracking has not been integrated into formal or legal enforcement mechanisms related to them. This is being pursued by conservation NGOs with interests in these MPAs, but the government has not prioritized funding or activities to support this application as yet. Discussion of this has been added in lines 261-5

Lines 205-207 indicate the fishery footprint data are most useful for coastal zone planning, are they not used in the current fisheries management practices?

To date the spatial information from PeskAAS has not been used to zone coastal areas by user groups, but protecting the rights and fishing ground of small-scale fishers is a key element of the new National fisheries strategy that PeskAAS helped to inform. There are some marine protected areas, but management is very limited and currently PDS tracking has not been integrated into enforcement mechanisms related to them. This discussion has been added in lines 256-268

Figures 2 and 3 are not cited in the text from what I can tell.

There was an error with the original file formatting as the section “Catch documentation at landings sites” is a methods section, not a figure caption. We have added “METHODS” at the start of the section to clarify the paper structure. Figure 3 is referenced at the start of this paragraph (originally line 95). Figure 2 is referenced just before the figure itself (originally line 87).

It is not clear what Supplementary Information 2 is supporting

Species were grouped to streamline the options available for enumerators, with the most important species for the fishery left as single species. This has an effect on what further analyses and statistical assessments can be carried out at species level (e.g. length frequency analysis for stock assessment of single species), so we considered it worthwhile to provide details. Supplementary Information 2 details how these species were categorised.

This has been further explained in lines 132-136: “Species of particular local importance were selected based on index of relative importance analysis on Timor-Leste fisheries and these are retained at species level. All other species captured in the fishery are classified by family, or aggregated further into one of the nine groups of species constituting the Marine Fishes division of the FAO International Standard Statistical Classification of Aquatic Animals and Plants (Supplementary information 1).”

Reviewer #3: 

Thank you very much for an interesting paper describing this new system for an open-source monitoring and analytics system. I agree that especially for small-scale fisheries it is a great problem to have reliable landings and catch data as a very important first step to establish some kind of fisheries management.

I think your system is well developed and you present the results in an appropriate fashion supported by the data. The results show that this could be a very useful tool for collecting data for fisheries managers. I have, however, some doubts how this system could be established in practice in low income countries. For the development of your system and to show how it works you have selected Timor-Leste as an example. For this trial you have organised the data collection by hiring data collectors (at least that is how it is described) and I am asking myself how this should work in the long run. Should the fishers report the data themselves (what I assume from your descriptions)? What will be the incentives of the fishers to participate in such a system? Misreporting is a problem in all fisheries management systems around the world.

Enumerators hired as part of the project who performed satisfactorily were taken on as government hired enumerators. These enumerators collect data from fishers at landing sites. At present, there is no fisher-submitted data in PeskAAS, nor does PeskAAS put the data into the hands of the fishers yet. The information does not currently bring new opportunities or benefits to individual fisheries actors other than long-term and relatively intangible assurances of resource sustainability. However, this is the aim of the next phase of PeskAAS development. This explanation of limitations and potentials has been integrated into the discussion (lines 269-301).

Therefore, could you describe a bit better how you foresee that this system will be used afterwards, who needs to participate in the system and what the authorities can do with this information. As I am not a fisheries biologist, I am also not sure about the fishbase database. Is the catch weight (average of all fishes in the catch, the single fish weights?) really the only parameter you need to judge the conditions of the fish stocks in mixed fisheries?

Length-weight parameters from FishBase are only used to calculate the weight of the particular family group in the catch according to the fish length. CPUE is a measure of relative abundance. This is not sufficient to carry out stock assessment or evaluate the stock health, except in relative terms (so compared to previously, or e.g. compared to another area or gear). However, it should be made clear that the analytics that are currently part of the PeskAAS online site are just a fraction of what the data can be used for – as you infer, much more is required to adequately manage fisheries. PeskAAS is the pipeline that joins all of the elements and components together, but there are many more packages, scripts, add-ons that can and will be developed for greater insight into particular fisheries or gears for decision making. This has been further clarified in the discussion in lines 261-301)

Line 36 ff.: I agree that it is a problem in low-income countries to collect data on catch/landings but the question is if those data alone would help to overcome the malnutrition in many countries due to the ability to establish a management system. I think we are still far away from introducing a fisheries management system with catch and/or landings data alone. Especially for mixed fisheries I believe that another approach could work. Many coastal communities have established some kind of management strategies as they have usually a substantial knowledge about the resources and have established some kind of self-management (e.g. in a community-based-management system). Is there information on self-management in the fishing communities in Timor-Leste?

We agree completely. Data alone will certainly not bring about better management. However, better management is almost impossible without data. Knowing the level of production of any resource is an absolute minimum piece of information if you are trying to manage its harvest adaptively over time. The process of setting objectives and making decisions based on the data is substantially more complex, but the development of co-management is the primary focus of the new national fisheries strategy developed in partnership with WorldFish, and recent work focused on highlighting potential opportunities and risks of co management and CBRM in Timor. We have added briefly to this section and more so to the discussion (lines 302-11) to highlight this point and the broader programmatic focus.

Lines 154 ff.: How does e.g. the number of fishers on a vessel influence the input data (fisher-hours)? The CPUE is very much depending on vessels size, fishing gear employed, number of fishers on board, etc. In small-scale fisheries often the owner alone is fishing or only a small number of fishers fish together.

We are not entirely clear on what is being asked by the reviewer, but there are two key reasons for collecting fisher numbers: 

1) We standardize effort into the unit of fisher-hours on each fishing trip, calculated as the trip duration in hours multiplied by the number of fishers onboard. The complexity of more robust CPUE standardization of multiple gears in tropical, multiple habitat, mixed species fisheries make it impractical in a data-deficient, low capacity context that is focused on providing data rapidly for livelihoods and nutrition interventions. Presuming all fishers on board are actively fishing, if there are 2 fishers onboard a vessel it will be fishing with twice the effort of a boat with 1 fisher. (“A task shared is a task halved”)

2) The catch survey question details the number of men, women and children fishing on the boat, which allows for the collection of gender-disaggregated data. This is a simple but crucial first step in beginning to understand more the contribution of women and children in fisheries, in order to bring great gender equality into decision-making processes and policy. 

Lines 166 ff.: I cannot judge how good those estimated numbers are. My guess is, however, that there are so many factors influencing catch levels that this may squeeze everything too much into a certain routine calculation. However, I am aware that this may be the best methodology available in this case.

No clear question or clarification, so no changes made to ms. However, the COVID19 pandemic has been an interesting example of just how accurately the geospatial tracking can illustrate drops in overall effort, and how this can be automatically tracked to decreases in fish production from catch data. 

Lines 183-185: Looks good but still I have doubts about the overall effort necessary to get all this detailed information and if there are no other possibilities.

No clear question asked, so no changes made to ms.

---

## [Decision Letter · Decision Letter 1]

27 Oct 2020

PeskAAS: A near-real-time, open-source monitoring and analytics system for small-scale fisheries

PONE-D-20-16520R1

Dear Dr. Tilley,

We’re pleased to inform you that your manuscript has been judged scientifically suitable for publication and will be formally accepted for publication once it meets all outstanding technical requirements.

Kind regards,

Ismael Aaron Kimirei, Ph.D.

Academic Editor

PLOS ONE

Additional Editor Comments (optional):

Reviewers' comments:

Reviewer's Responses to Questions

**Comments to the Author**

1. If the authors have adequately addressed your comments raised in a previous round of review and you feel that this manuscript is now acceptable for publication, you may indicate that here to bypass the “Comments to the Author” section, enter your conflict of interest statement in the “Confidential to Editor” section, and submit your "Accept" recommendation.

Reviewer #1: All comments have been addressed

Reviewer #2: (No Response)

Reviewer #3: All comments have been addressed

2. Is the manuscript technically sound, and do the data support the conclusions?

Reviewer #1: Yes

Reviewer #2: Yes

Reviewer #3: Yes

3. Has the statistical analysis been performed appropriately and rigorously? 

Reviewer #1: Yes

Reviewer #2: Yes

Reviewer #3: Yes

4. Have the authors made all data underlying the findings in their manuscript fully available?

Reviewer #1: Yes

Reviewer #2: Yes

Reviewer #3: Yes

5. Is the manuscript presented in an intelligible fashion and written in standard English?

Reviewer #1: Yes

Reviewer #2: Yes

Reviewer #3: Yes

6. Review Comments to the Author

Reviewer #1: (No Response)

Reviewer #2: I would like to thank the authors for their responsiveness and extensive effort in addressing reviewer comments. The data flow is much clearer now and the authors response to Reviewer 2 regarding the different spatial scales of the data collected was extremely helpful, although not necessary the authors could consider adding this explicit description to the abstract or overview. I particularly like the last sentence of the abstract and think it highlights nicely how information from this tool could be useful. I recommend publishing this article as a nice description of the authors analytical tool to collate fisheries data. I would also like to encourage the authors to consider another publication highlighting lessons learned from the full process they undertook in Timor-Leste. For instance, the development of the fisheries form sounds like it took an extensive collaborative effort with the Timorese government and fishers that could be beneficial to share with those interested in utilizing this approach in other countries and circumstances.

Below are a few grammatical edits to consider before publication.

Line 13: insert ‘the’ before world’s fish catch

Line 40: remove the ‘s’ from nutrients

Line 100: should be ‘catering to humanitarian organizations’ instead of ‘for’

Line 130: could consider expanding on what these various feedback mechanisms are

Line 153: the dash (-) is noted needed between interface and that

Line 222: function nn2 and package RANN are in a different font, the R package on line 156 is not in a different font

Line 223: is 0.20 km, there isn’t a unit of distance here

Line 224: should read available as of July (change the at to of)

Line 240: I would recommend changing sustainability to sustainable fishing

Line 244: add (PDS) after Pelagic Data Systems

Line 280: envisage should be envision

Reviewer #3: Dear Authors,

Thank you for your revised manuscript. I have no further comments although I would still like to know if in Timor fishers implemented something like community-based-management systems to limit fishing effort outside of governmental intervention. However, I am aware that this goes to far for the paper.

7. PLOS authors have the option to publish the peer review history of their article (what does this mean?). If published, this will include your full peer review and any attached files.

Reviewer #1: No

Reviewer #2: No

Reviewer #3: No

---

## [Editor Report · Acceptance letter]

30 Oct 2020

PONE-D-20-16520R1 

PeskAAS: A near-real-time, open-source monitoring and analytics system for small-scale fisheries 

Dear Dr. Tilley:

I'm pleased to inform you that your manuscript has been deemed suitable for publication in PLOS ONE. Congratulations! Your manuscript is now with our production department. 

Kind regards, 

on behalf of

Dr. Ismael Aaron Kimirei 

Academic Editor

PLOS ONE